# Timing of Vertical Head, Withers and Pelvis Movements Relative to the Footfalls in Different Equine Gaits and Breeds

**DOI:** 10.3390/ani12213053

**Published:** 2022-11-07

**Authors:** Marie Rhodin, Ineke H. Smit, Emma Persson-Sjodin, Thilo Pfau, Vikingur Gunnarsson, Sigridur Björnsdóttir, Ebba Zetterberg, Hilary M. Clayton, Sarah J. Hobbs, Filipe Serra Bragança, Elin Hernlund

**Affiliations:** 1Department of Anatomy Physiology and Biochemistry, Swedish University of Agricultural Sciences, S-750 07 Uppsala, Sweden; 2Department of Equine Sciences, Faculty of Veterinary Medicine, Utrecht University, Yalelaan 112-114, NL-3584 CM Utrecht, The Netherlands; 3Faculty of Kinesiology, University of Calgary, Calgary, AB T2N 1N4, Canada; 4Faculty of Veterinary Medicine, University of Calgary, Calgary, AB T2N 1N4, Canada; 5Equine Science Department, Hólar University, 551 Saudárkrókur, Iceland; 6Faculty of Agricultural Sciences, Agricultural University of Iceland, 311 Hvanneyri, Iceland; 7Sport Horse Science, 3145 Sandhill Road, Mason, MI 48854, USA; 8Research Centre for Applied Sport, Physical Activity and Performance, University of Central Lancashire, Preston PR1 2HE, UK

**Keywords:** inertial measurement units, gait, biomechanics, objective motion analysis, lameness

## Abstract

**Simple Summary:**

Movement symmetry of the head and pelvis are used to measure lameness in horses in trot. Although head, pelvis and limb movements have been described, less is known about the temporal relationships between them. This information is needed to understand how the movements change with lameness. This is particularly relevant in gaited horses, such as the Icelandic horse that perform gaits such as tölt and pace, which are challenging to evaluate. This study used inertial measurement units to investigate head, withers and pelvis motion relative to limb movements in Icelandic, Warmblood and Iberian horses. Limb movements, together with vertical movements and lowest/highest positions of the head, withers and pelvis were calculated, and the relative timing of the events was compared across breeds. Additionally, data for tölt and pace were collected and evaluated in ridden Icelandic horses. For all gaits except walk and pace, the lowest/highest positions of the head/withers/pelvis were closely temporally related to midstance and hoof-off, respectively. Pelvic and withers total range of motion differed between all breeds. The Icelandic horses showed shorter stride duration and smaller movements of the upper body than the other breeds at trot, which may explain why lameness evaluation in this breed is challenging.

**Abstract:**

Knowledge of vertical motion patterns of the axial body segments is a prerequisite for the development of algorithms used in automated detection of lameness. To date, the focus has been on the trot. This study investigates the temporal synchronization between vertical motion of the axial body segments with limb kinematic events in walk and trot across three popular types of sport horses (19 Warmbloods, 23 Iberians, 26 Icelandics) that are known to have different stride kinematics, and it presents novel data describing vertical motion of the axial body segments in tölting and pacing Icelandic horses. Inertial measurement unit sensors recorded limb kinematics, vertical motion of the axial body at all symmetrical gaits that the horse could perform (walk, trot, tölt, pace). Limb kinematics, vertical range of motion and lowest/highest positions of the head, withers and pelvis were calculated. For all gaits except walk and pace, lowest/highest positions of the pelvis and withers were found to be closely related temporally to midstance and start of suspension of the hind/fore quarter, respectively. There were differences in pelvic/withers range of motion between all breeds where the Icelandic horses showed the smallest motion, which may explain why lameness evaluation in this breed is challenging.

## 1. Introduction

An understanding of gait mechanics and normal upper body movement patterns is an essential prerequisite both for visual evaluation and for the development of algorithms to detect lameness based on the presence of movement asymmetries in symmetrical gaits. 

Automated lameness detection is based on asymmetrical movement of the axial body segments in symmetrical gaits, which are affected by limb kinematics, ground reaction forces, and whether the gaits use inverted pendulum or spring mass mechanics to conserve energy in the stance phase [1]. These mechanisms will be explained further in the sections on the specific gaits. Previous studies have provided information on breed-specific differences in limb kinematics and kinetics [2], but relatively little is known about the magnitude, range, and timing of movements of the axial body segments.

The present study extends previous work by comparing between-breed reference values for axial body segment kinematics in sound Warmblood, Iberian, and Icelandic horses at walk and trot in hand. Icelandic horses are classified as gaited horses, which implies the ability to perform additional gaits beyond the typical walk, trot and canter/gallop. The amble is a blanket term covering a large number of four-beat gaits performed by gaited horses that have the same footfall sequence as walk but vary in footfall timings and speed of progression. A horse’s ability to amble and to pace is largely genetically determined by a nonsense mutation in the DMRT3 gene, also known as the “gait-keeper” gene [3,4]. Icelandic horses that are heterozygous for the DMRT3 mutation are able to tölt, and those that are homozygous can also pace [3]. These gaits are difficult to evaluate using methods applied to walk and trot, and the relatively fast speeds at which they are performed dictate that they are usually evaluated under saddle. Therefore, ridden data for Icelandic horses is presented in order to cover the spectrum of symmetrical gaits in this breed.

Objective evaluation of lameness augments the clinical gait examination by detecting asymmetries that are below the threshold of detection by the human eye [5] while avoiding inherent observer bias [6]. Trot has been the focus of objective lameness evaluation studies due to its symmetrical nature and the higher loading of the limbs compared to walk [7]. A few studies have objectively described the kinematic and kinetic changes in lame horses at walk [7,8], but an attempt to detect lameness objectively during gallop proved unsuccessful [9].

Walk is a four-beat gait with a lateral sequence and successive footfalls separated by approximately 25% of stride duration. As in all symmetrical gaits, the movements of the contralateral limb pairs are out of phase, and there is only a short period of overlap between contact of one limb and lift off of the contralateral limb. The walk uses inverted pendulum dynamics so that the withers and croup are low at hoof-on, vault over the grounded hoof to their highest position during midstance [7], then descend again to lift-off [10,11]. The withers and croup are raised and lowered alternately, with out-of-phase oscillations twice per stride [7]. Vertical head movements are out of phase with the withers and in phase with the croup [7].

Trot is a two-beat, diagonally coordinated gait with a sinusoidal centre of mass trajectory that is typical of spring-mass mechanics. It incorporates two suspension phases per stride that separate the lift off of one limb from the contact of the contralateral limb so that there is no overlap between them. The trot uses spring-mass mechanics in which the withers and croup are high at the start of stance, descend as the joints flex and limb length shortens in midstance, and rise again at the end of stance [1,11]. The head, withers and croup rise and fall synchronously twice per stride, with minima during the diagonal stance phases and maxima around the time of push-off into the suspension phase [8]. However, the exact timing and variations of these movements have not yet been fully described. Breed differences in limb kinematics of trot have been reported. For example, Iberian horses have a shorter stride length, higher stride frequency and lower dorsoventral displacement and activity than Warmbloods at trot [2], which is likely to affect vertical motion of the axial body segments.

Tölt is a four-beat, lateral sequence, ambling gait without suspension. It has eight limb support phases normally alternating between bipedal and unipedal limb support. Ideally, tölt should have a regular four-beat rhythm with 25% of stride duration elapsing between hoof placements [12]. Due to the frequent periods of single limb support, it has been suggested that it is more difficult for horses to mask lameness in tölt than trot [13]. The mechanical characteristics of tölt are more similar to those of a running gait using spring-mass mechanics but the footfall patterns, and small vertical excursions of the centre of mass are typical of a walking gait [8]. Pattern recognition methods indicated that tölt is more aligned to running than walking gaits [1]. Little is known about relative timing of the head, withers and croup oscillations in tölt.

Pace is a two-beat, laterally synchronised gait with two suspension phases per stride and is classified as a running gait [12]. At high speed, the pace becomes four-beat with hind-first lateral dissociation [14]. Pacing is most often performed by harness-racing horses, and Icelandic horses are somewhat unusual in that they are ridden at pace. Little is known about the timing of the head, withers and croup oscillations in pace.

Differentiation between sound and lame horses has, to date, been based primarily on differences in magnitude of the two minima and maxima of the axial body segments at trot. Differences in head oscillations are used to evaluate forelimb lameness, while differences in croup oscillations are used to evaluate hindlimb lameness [8,15,16,17,18]. In the hind limbs, differences in vertical displacement between the two minima (PDmin) indicate an impact lameness during the first half of stance in which weight acceptance and, therefore, lowering of the body, are reduced on the lame side. Differences between the two maxima (PDmax) indicate a push-off lameness during the second half of stance that propels the pelvis to a lower maximum position after lame limb stance [19]. Vertical withers movements differentiate a primary forelimb lameness from a compensatory head nod shown by horses with a primary hindlimb lameness [20] based on relative timing (phase shift) between head and withers maxima [21]. Information is needed describing the variation in vertical range of motion (ROMz) and timing of the maxima and minima of the head, withers and pelvis in sound horses of different breeds not only at the trot but also at other symmetrical gaits. This information is fundamental to understanding lameness mechanics in different gaits and types of horses.

The first objective is to compare the range and timing of vertical motion of the head, withers, and pelvis in sound Warmblood, Iberian and Icelandic horses performing walk and trot in hand. The second objective is to present key kinematic differences between Warmblood, Iberian and Icelandic horses that may affect visual lameness assessment. The third objective is to present reference data describing the range and timing of vertical movements of the head, withers and croup that are specific to Icelandic horses ridden in walk, trot, tölt and pace.

Across each gait, we hypothesized that (1) absolute timing of limb kinematic variables would vary between breeds; (2) absolute differences in ROMz at all axial locations would be consistently larger in Warmbloods compared to Iberians and larger in Iberians compared to Icelandic horses, not only due to the discrepancy in size but also associated with breed-specific gait characteristics; (3) the relative timing of minima and maxima within strides would not vary between breeds.

## 2. Materials and Methods

### 2.1. Study Design and Study Population

Horses included in this observational study were from previously collected data sets from horses that were sound according to the owner and with only minor objectively measured movement asymmetries that were below predefined thresholds (see the section “Data processing and analysis”) when trotting on the straight (Figure 1).

The final data set consisted of horses that were in training at various levels; some were competing and some were leisure horses. Nineteen Warmblood riding horses (11 mares, 8 geldings, age 10 ± 4.2 years, withers height 165.5 ± 7.0 cm) performed 1340 (range per horse 19–202) strides at trot and 1000 (range per horse 19–126) strides at walk in-hand.

Twenty-three Iberian horses (10 Lusitano, 13 Pura raza Española, 6 mares, 13 geldings, 4 stallions, age 9 ± 2.5 years, withers height 164.6 ± 4.9 cm) performed 1540 (range per horse 46–119) strides at trot and 1090 (range per horse 19–147) strides at walk in-hand.

Twenty-three Icelandic horses (10 mares, 13 geldings, age 10 ± 3.7 years, withers height 141 ± 3.7 cm) performed a total of 815 (range per horse 34–145) strides at trot and 837 (range per horse 27–98) strides at walk in-hand. When ridden 23 horses performed 1920 strides at trot (range per horse 28–203), and 2020 strides at walk (range per horse 25–424), 11 horses performed 410 (range per horse 6–82) strides at pace and 18 horses performed 3100 (range per horse 19–484) strides at tölt.

### 2.2. Data Sets

Data were collected in five different research projects between 2017 and 2022 in Sweden, Iceland, the United States of America and the Netherlands, using data from seven IMU sensors (EquiMoves^®^, The Netherlands) [22] per horse, with a sampling frequency of 500 Hz for measurements in data sets 1–2 and 200 Hz for measurements in data sets 3–5 (Figure 1). The low-g accelerometer in the IMU sensors was set at ±16 g, the high-g accelerometer was set at ±200 g, and the angular velocity was set at 2000 degrees/s. The sensors were attached dorsally to the poll, withers and pelvis in between *Tubera sacrale*, and to the lateral aspect of each metapodial segment attached on brushing boots. Since the horses were accustomed to wearing brushing boots, they were not disturbed by the limb sensors and none of them reacted to the upper body sensors. Each horse had at least 5 min for acclimatization before measurements started. All horses were measured during walk and trot in hand with different handlers, and the Icelandic horses were also measured at ridden walk, trot, tölt and pace in a straight line on a hard surface (lava sand and packed dirt (Figure 1)).

### 2.3. Data Collection and Selection

Sampled data were transmitted via the gateway to the computer. After each measurement, missing samples on the computer that were not transmitted during live measurement were downloaded from the memory card in the sensors. The software (EquiMoves^®^) stored the data and pre-computed orientation in log files on the computer. All data retrieved by the software were stored and logged for post-analysis. The data collection was synchronized with video recordings for later data quality checks. The data processing procedure and inclusion criteria are depicted in Figure 1. Algorithms developed previously by machine learning, with an accuracy threshold set to 80%, were used to classify the different gaits [23] and the segments of walk, trot, tölt and pace performed in a straight line were selected for further data analysis. The data were processed using Matlab 2020b (Mathworks, Natick, MA, USA).

### 2.4. Calculated Parameters

For each segment, the variables of interest were calculated, starting by detection of the moments of hoof-on/off for each limb using the data from sensors placed laterally on the metapodial segment of each limb [24] as previously described [22]. Once limb timings had been correctly identified, upper body accelerometer data were double integrated to provide displacement data using previously published methods [25]. Based on the time of left-hind hoof contact, the vertical displacement output of the head, withers and pelvis sensors were segmented into strides and normalized to stride duration, where a stride was defined as the time between consecutive hoof contacts of the left hind limb. Stance normalization was performed based on left front/hind contact and lift-off, respectively. Normalization was set to 100% to enable comparisons between horses with different stride/stance times.

The lowest and highest positions of the head (Hmin/Hmax), withers (Wmin/Wmax) and pelvis (Pmin/Pmax) were calculated from the sensors on the poll, withers and between the tubera sacrale (croup) for each stride, as previously described [22]. Differences between the two minima and between the two maxima for the head (HDmin, HDmax) and pelvis (PDmin, PDmax) were calculated, to determine symmetry during in-hand trot. Horses were excluded if the absolute HDmin and/or HDmax values exceeded 16 mm, and/or if PDmin and PDmax exceeded 8 mm at straight line trot to avoid using horses with marked asymmetry or an underlying lameness.

Diagonal dissociation (time between hoof contacts of the diagonal limbs) was calculated for walk and trot and lateral dissociation (time between hoof contacts of the ipsilateral limbs) was calculated for all gaits and as percentage of stride duration (%StrD). They were defined as positive if hind hoof contact preceded that of the fore hoof. Suspension time was calculated as percentage of stride duration (%StrD) during which none of the hooves were in contact with the ground. It was compared between breeds at trot. Duty factor, which expresses stance phase duration as a proportion of the stride duration, was also calculated.

### 2.5. Statistical Analysis

Results are presented as descriptive statistics. To test the effect of breed (Icelandic vs. Warmblood vs. Iberian) and anatomical location (head, withers and pelvis) on the target variables, linear mixed models were built in R-Studio (version 1.3, Boston, MA, USA). Three models were built: (1) to test the effect of breed and anatomical location on upper body ROMz using horse ID as random effect and the interaction between breed and anatomical location as fixed effect; (2) to test the effect of breed on timing of minima and maxima of head, withers and pelvis positions in relation to stance phase, using breed as fixed effect and horse ID as random effect; and (3) to test the effect of breed on limb kinematic parameters (suspension time, stride duration, diagonal dissociation) using horse ID as random effect and breed as fixed effect. Model results (for ROMz, see Figure 2) are presented as estimated marginal means (EMM) and lower/upper confidence intervals (CI) using the package “emmeans” (version 1.4.5, Boston, MA, USA). Significance was set at *p* < 0.05, and *p* values were adjusted for multiple comparison using the Tukey method. Model adequacy (normality and constancy of variance) was confirmed using visualisation of the scatter plot residuals vs. fitted values and explanatory variables, respectively, and QQ-plots. The remaining discrete variables are presented as median and interquartile range (IQR).

## 3. Results

### 3.1. Temporal Limb Data

Results of temporal limb measurements for the different breeds, gaits and ridden or in-hand conditions are presented in Table 1.

### 3.2. Stride Duration at Walk and Trot

Stride duration was significantly shorter (*p* < 0.001) for Icelandic horses at walk and trot than for Iberian and Warmblood horses. Stride duration was also significantly shorter for Iberian horses than for Warmblood horses at both walk (*p* < 0.03) and trot (*p* < 0.013). No significant differences were seen between in-hand and ridden conditions for the Icelandic horses at walk, but stride duration was significantly longer (*p* < 0.001) during in-hand trot compared with ridden trot. The model outcomes in the form of estimated marginal means are presented in Appendix A.

### 3.3. Suspension Time at Trot

There were no significant differences (*p* < 0.991) in suspension time as a percentage of stride duration between Warmblood horses and Icelandic horses. A trend (*p* = 0.06) towards a longer suspension time was seen for Warmblood horses compared with Iberian horses when trotting in hand. A significantly (*p* < 0.020) longer suspension time was seen for Icelandic horses compared with Iberian horses. For ridden Icelandic horses, suspension time was significantly (*p* < 0.001) shorter compared with in-hand trot. The model outcomes in the form of estimated marginal means are presented in Appendix A.

### 3.4. Diagonal Dissociation at Trot

There were significant differences (*p* < 0.001, *p* < 0.028) in diagonal dissociation in trot between Iberian and Warmblood horses, and between Iberian and Icelandic horses when trotting in hand. No differences (*p* < 0.339) were seen between Icelandic horses when ridden vs. in-hand trot. The model outcomes in the form of estimated marginal means are presented in Appendix A.

### 3.5. Vertical Head, Withers and Pelvic Range of Movement

At walk, pairwise breed comparison showed that the Icelandic horses had a smaller ROMz of the head compared to Warmblood and Iberian horses. No significant differences were found for withers and pelvis ROMz between the breeds (Table 2). All within-breed pairwise comparisons of ROMz of the three body segments (head, withers, pelvis) were significant. The head showed the largest ROMz, followed by the pelvis, while the withers showed the smallest.

At trot, Warmblood horses showed significantly larger head ROMz than Iberian and Icelandic horses, but the difference between Icelandic and Iberian horses was not significant. ROMz for withers and pelvis was highest in Warmbloods and lowest in Icelandics with significant differences between all breeds (Table 3). Within-breed comparison of the three body segments showed no differences in Iberian horses. In Icelandic horses, the withers had a lower ROMz than the head or pelvis, while in Warmblood horses, the head had a lower ROMz than the withers or pelvis (Table 3).

### 3.6. Timing of Vertical Movement of Head, Withers and Pelvis

Results of vertical displacement minima/maxima timing of the head, withers and pelvis in relation to the limb events at different gaits are presented in Figure 3, Figure 4, Figure 5 and Figure 6 and Table 4.

#### 3.6.1. Walk

Descriptive results of the timing of the minima and maxima of the axial body segments (head, withers, pelvis) in relation to the footfall patterns during walk for all horses within the three breeds are presented in Figure 3.

#### 3.6.2. Trot

Descriptive results of the timing of the minima and maxima of the axial body segments (head, withers, pelvis) in relation to the footfall patterns during trot for all horses within the three breeds are presented in Figure 4.

#### 3.6.3. Tölt

Descriptive results of the timing of the minima and maxima of the axial body segments (head, withers, pelvis) in relation to the footfall patterns during tölt for the Icelandic horses are presented in Figure 5.

#### 3.6.4. Pace

Descriptive results of the timing of the minima and maxima of the axial body segments (head, withers, pelvis) in relation to the footfall patterns during pace for the Icelandic horses are presented in Figure 6.

Differences in relative timing of minima and maxima between the different body segments (head–withers, head–pelvis and withers–pelvis) in the three breeds when performing different gaits are presented in Table 5 and Figure 7 and Figure 8.

## 4. Discussion

This study revealed clear differences not only in limb kinematics but also in upper body kinematics between Icelandic, Warmblood and Iberian horses. It also demonstrated that such differences can be quantified in field conditions using mobile sensor technology. These findings open the way for more kinematic studies involving between- and within-breed comparisons and represent a first step in determination of reference kinematic values. Modern objective lameness assessment methods make use of symmetry values to overcome the lack of absolute reference values [26]; hence, establishing reference values for specific breeds offers the possibility to further standardize gait analysis techniques in the near future, in order to establish breed-specific gait patterns. 

At trot, Warmblood horses showed a greater vertical range of motion of the upper body than Icelandic and Iberian horses, which might be expected due to producing relatively larger vertical impulses compared to other breeds [27]. Barrey et al. [2] also found a greater vertical displacement of sternum in Warmblood horses compared to Spanish horses. Our results clearly showed a close temporal relationship between vertical movements of the upper body and the limb stance phases, which is very consistent across breeds at a particular gait. Therefore, it is likely that upper body movement asymmetry, which is used for lameness quantification at trot, can also be a useful variable for lameness evaluation in symmetrical gaits other than trot [26]. 

### 4.1. Temporal Variables

#### 4.1.1. Stride Duration

Stride duration differed between all breeds at both walk and trot, with Icelandic horses showing the shortest stride duration in hand, although it increased when they were ridden. Warmblood showed the longest stride duration. There were no differences in relative suspension time between Warmblood and the other two breeds when trotted in hand, but Icelandic horses showed a longer suspension time compared to Iberian horses. The short stride duration in Icelandic horses can be related to their shorter limb length, lower body mass, and the consequently higher stride frequency. These characteristics may contribute to making visual lameness assessment in this breed more challenging [28]. 

#### 4.1.2. Diagonal Dissociation at Trot

On average, horses in the present study showed forelimb-first diagonal dissociation at trot, which agrees with earlier studies on Icelandic horses [29] and Warmblood horses [30] trotting in hand on a treadmill. In Iberian horses, the fore-first diagonal dissociation at trot was shorter than in Warmblood horses and Icelandic horses. Earlier landing of the hind hoof compared with the diagonal forelimb has been observed in previous studies on Warmblood horses and has been associated with higher trot quality [31]. It has also been reported in ridden Warmblood horses performing collected, working, medium and extended trot that diagonal dissociation was shortest during collection [32]. The fore-first dissociation in Icelandic horses may be associated with their lower quality of trot compared with the other breeds and may also be affected by the DMRT3 gene mutation seen in Icelandic horses [3,4]. 

#### 4.1.3. Relative Time Shifts between Segments

The vertical minima (Hmin/Wmin/Pmin) occurred around midstance for all gaits except for pace (Hmin/Wmin around 27% of stance) and for Pmin and Wmin at walk, where the lowest positions were reached during dual hindlimb and dual forelimb support, respectively, when the contralaterally limbs were maximally spread apart. All Hmax/Wmax/Pmax occurred within the period from the last 15% of stance to the first 10% of the next limb stance phase, except for Pmax and Wmax at walk (42–56% of stance phase). Therefore, changes in vertical movement symmetry for Hmin/Wmin/Pmin are probably good indicators of weight-bearing lameness in tölt and pace. PDmax is likely to be a good indicator of push-off lameness [19] except in walk where the pelvis reaches its highest position during mid-stance. In horses with weight-bearing forelimb lameness during walk, adaptive movement strategies appearing as differences in the lowest position of the head (HDmin) and withers (WDmin) have been reported [7]. Further studies are needed in hind limb lame horses during walk and in lame horses in gaits other than trot to identify their compensatory lameness strategies, and thereby improve subjective and objective lameness detection.

Timings of vertical minima and maxima of head, withers and pelvis were synchronised for trot and pace. For walk, the maxima and minima of the head and sacrum were in phase, while the withers were around 25% out of phase compared with both head and sacrum. For tölt, timings of the minima and maxima of head and withers were synchronised, while pelvis minima and maxima were around 25% out of phase with head and withers.

### 4.2. Vertical Head, Withers and Pelvis Range of Movement

Different approaches are currently used for lameness quantification. For example, the differences between minima and maxima are used for head (HDmin/HDmax), withers (WDmin/WDmax) and pelvis (PDmin/PDmax) (EquiGait^®^, EquiMoves^®^, Q-horse^®^) or these values are corrected for the amplitude of the second harmonic (Equinosis^®^) [33]. The lower range of motion of the head observed in this study for Icelandic and Iberian horses compared with Warmblood horses may affect the magnitude of these asymmetry variables differently, but we only know how the range of motion is affected by lameness in Warmblood horses [7]. The range of motion of head, withers and pelvis is also negatively correlated to speed [34], as speed increases the range of motion of head, withers and pelvis decreases. Speed has less influence on movement symmetry variables during straight line trot [28], but if these variables are corrected for vertical range of motion, they will probably also be affected by speed [28]. This is also very relevant when performing visual lameness evaluation, which is based largely on assessment of upper body movement symmetry. If the ROMz is smaller, as observed here for Icelandic horses at trot and for Iberian horses, the capacity of the human eye to detect this motion might be hampered [5], making it more challenging to detect lameness at trot in the Icelandic or Iberian horse compared with the Warmblood horse.

### 4.3. Implications for Lameness Assessment of Different Breeds

A short stride duration, i.e., a high stride frequency, may be one of the factors making visual lameness assessment more challenging [28]. This, in combination with a smaller ROMz at trot, might call into question the accuracy of visual lameness assessment for Icelandic horses, supporting the use of gait analysis methods in this breed. In the present study, Icelandic horses showed a smaller ROMz of the head at walk and of the head, withers and pelvis at trot, which in combination with the higher stride frequency, may also partly explain why lameness evaluation in this breed is regarded as very challenging. In addition, there may be differences in lameness adaptation strategies between breeds due to the DMRT3 gene mutation, which may allow Icelandic horses to alter their footfall pattern more easily than Warmblood horses. Therefore, further studies are needed on lameness compensation in gaited horses.

### 4.4. Inclusion Criteria

For all data sets, the initial inclusion criterion was “sound horse in training”, but despite this, many horses presented with a high degree of motion asymmetry as seen in previous studies [35]. Therefore, an additional selection was made for the present study to avoid the influence of obvious lameness on the kinematic variables of interest. The thresholds chosen were slightly above the reference literature on inter-run variability in sound warmblood horses [36,37] but within Thoroughbred repeatability values [21], taking into account the substantial systematic differences between measuring systems [38]. Data were also selected by using previously developed gait classification algorithms [23] for the different gaits. The neural network algorithm combines several of the parameters tested to make a decision regarding whether a particular stride is classified as a specific gait (walk, trot, tölt or pace). For this study, we set a classification threshold of 80%, meaning that any strides used by the combination of the selected variables to predict the gait [23] which yielded a classification with accuracy <80% were removed. This could have affected our results by a selection bias towards only including strides where horses displayed features with greater weight in the classification, with those horses being overrepresented in the data set. Classification of tölt and pace based on visual labelling of strides from video [23] can also have led to biased selection of specific features that were only selected initially from video observation. Based on the 80% accuracy thresholds, no gaits from the horses were excluded from the final results due to gait classification, meaning that all 26 Icelandic, 19 Warmblood and 23 Iberian horses are represented in the results. 

### 4.5. Limitations

This study includes a limited number of horses per breed. Consequently, factors that could have been interesting to study in relation to the presented data, such as conformation, training level and size, could not be included in our analysis. Due to the multi-centre nature of this study, differences in surface between and within breeds can be expected and might have affected some of the stride temporal variables measured at the different sites [39]. In addition, there is some potential clustering effect within breeds, given that some horses are from the same locations. At the same time, within each breed, data were collected in different countries. Not all data sets contained speed that could be used in the statistical models, and therefore, speed was not taken into account during statistical analysis [30]. The proximal limb angles may influence the stride variables but were not measured in the present study [40].

## 5. Conclusions

Vertical minimum and maximum position of head, withers and pelvis generally occurred around midstance of the forelimb and hindlimb for all gaits except for head and withers in pace and pelvis and withers at walk. The highest vertical position of head, withers and pelvis occurred from the last 15% of stance through the first 10% of the following limb stance phase, except for the withers and pelvis at walk. Icelandic horses showed a smaller vertical range of motion of the head at walk and of the head, withers and pelvis at trot. This, in combination with higher stride frequency, may partly explain why lameness evaluation in this breed is regarded as very challenging.

## Figures and Tables

**Figure 1 animals-12-03053-f001:**
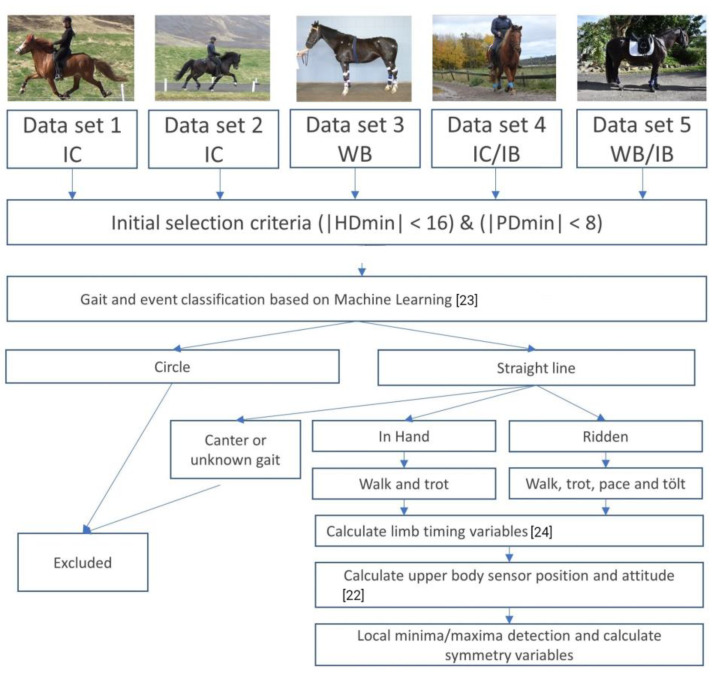
Schematic representation of the data-processing procedure applied for the different data sets. IC: Icelandic horse, WB: Warmblood horse, IB: Iberian horse. HDmin: difference in minimum position of the head between left and right diagonals at trot. PDmin: difference in minimum position of the pelvis between left and right diagonals at trot.

**Figure 2 animals-12-03053-f002:**
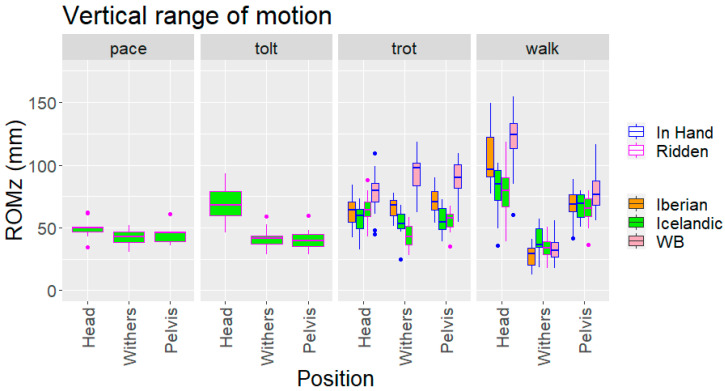
Vertical range of motion (ROMz) of upper body segments for different gaits, breeds and conditions (ridden vs. in hand). WB: Warmblood horses.

**Figure 3 animals-12-03053-f003:**
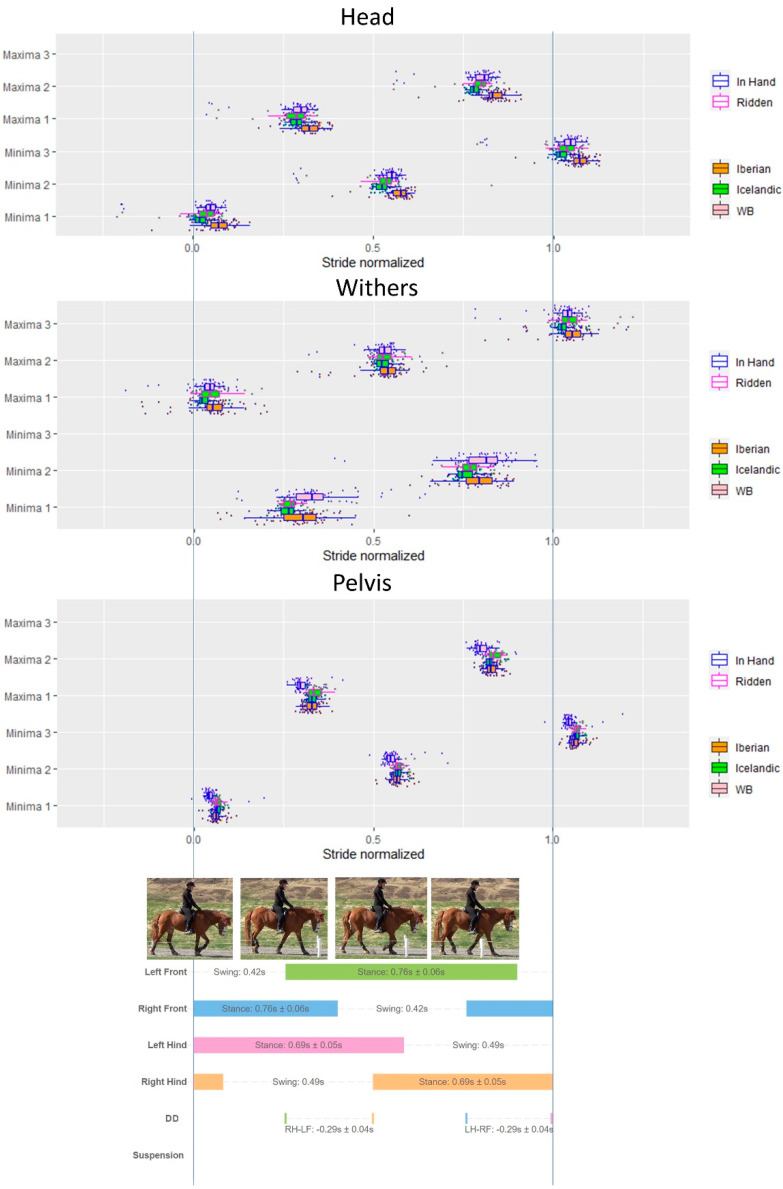
Combined data for the three breeds (Iberian, Icelandic, Warmblood) to show timing of minima (Min) and maxima (Max) on *y*-axis of the head, withers and pelvis during a normalized walk stride, including data from in hand and ridden conditions. Each dot represents a stride. Stance bars for each limb indicate mean and standard deviation (SD) of all strides across breeds and conditions. Diagonal dissociation (DA) and SD for right hindlimb (RH)–left forelimb (LF) diagonal and left hindlimb (LH)–right forelimb (RF) diagonal are presented.

**Figure 4 animals-12-03053-f004:**
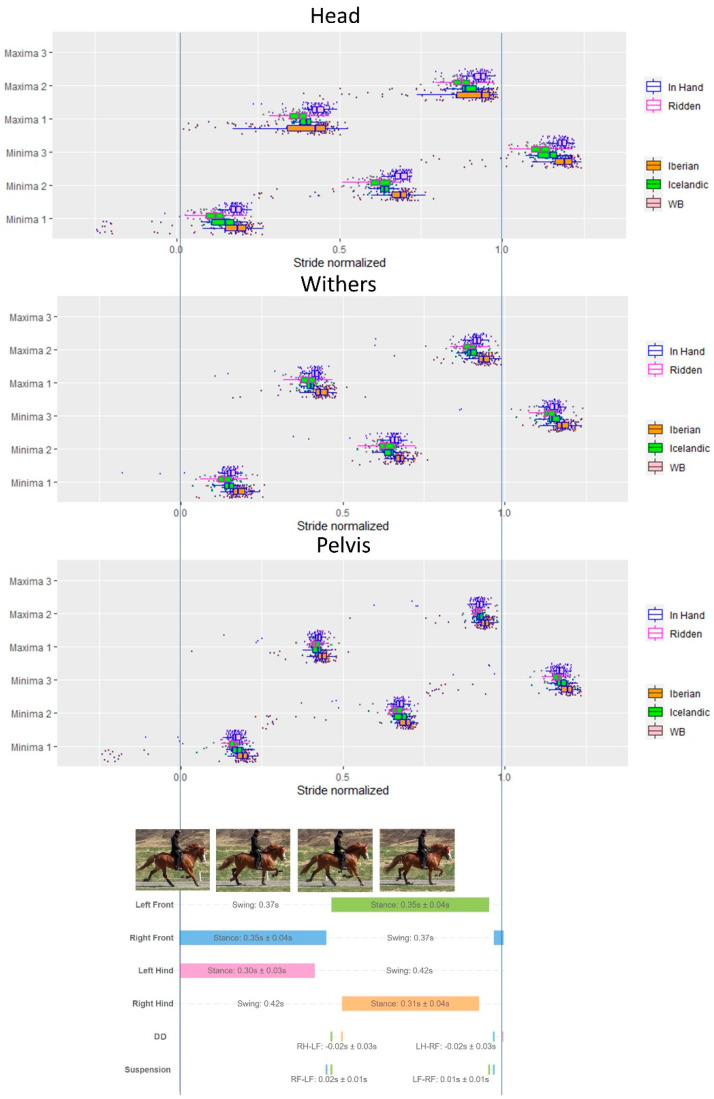
Combined data for the three breeds (Iberian, Icelandic, Warmblood) to show timing of minima (Min) and maxima (Max) on *y*-axis of the head, withers and pelvis during a normalized trot stride, including data from in hand and ridden conditions. Each dot represents a stride. Stance bars for each limb indicate mean and standard deviation (SD) of all strides across breeds and conditions. Diagonal dissociation (DD) and SD for right hindlimb (RH)–left forelimb (LF) diagonal and left hindlimb (LH)–right forelimb (RF) diagonal are presented. Suspension time is also presented.

**Figure 5 animals-12-03053-f005:**
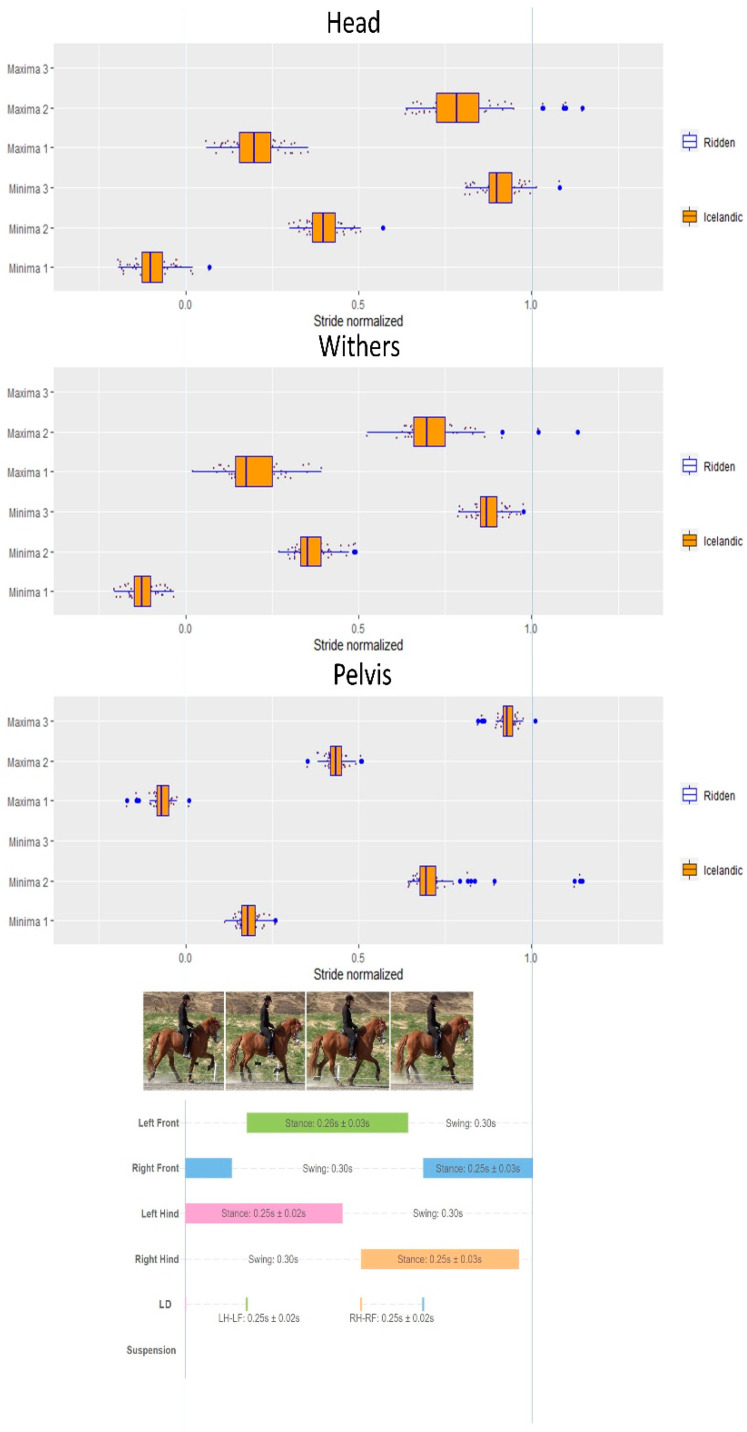
Timing of minima and maxima (*y*-axis) of the head, withers and pelvis during a normalized tölt stride (0–1.0) for the ridden Icelandic horses. Each dot represents a stride. Stance bars for each limb indicate mean and standard deviation (SD) of all strides. Lateral dissociation (LD) and SD for left hindlimb (LH)–left forelimb (LF) and right hindlimb (RH)–right forelimb (RF) are presented.

**Figure 6 animals-12-03053-f006:**
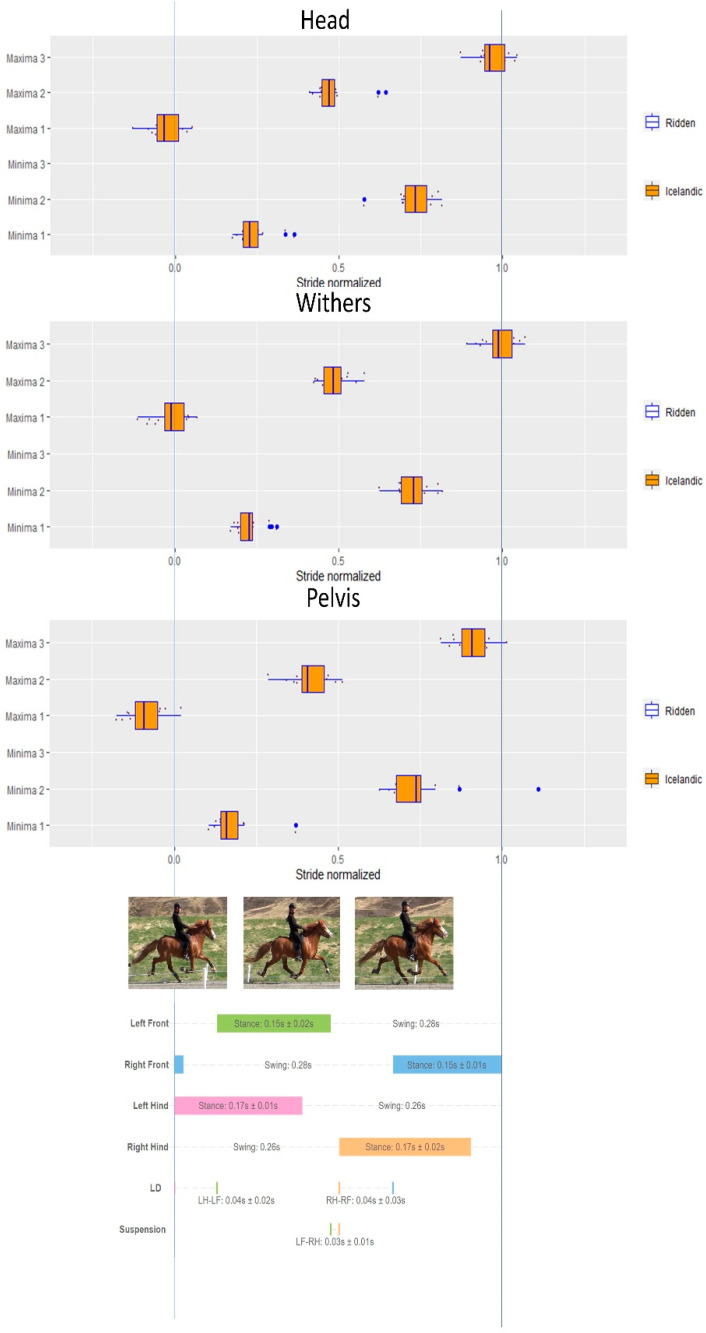
Timing of minima and maxima (*y*-axis) of the head, withers and pelvis during a normalized pace stride (0–1.0) for the ridden Icelandic horses. Stance bars for each limb indicate mean and standard deviation (SD) of all strides. Lateral dissociation (LD) and SD for left hindlimb (LH)–left forelimb (LF) and right hindlimb (RH)–right forelimb (RF) are presented. Suspension time is also presented.

**Figure 7 animals-12-03053-f007:**
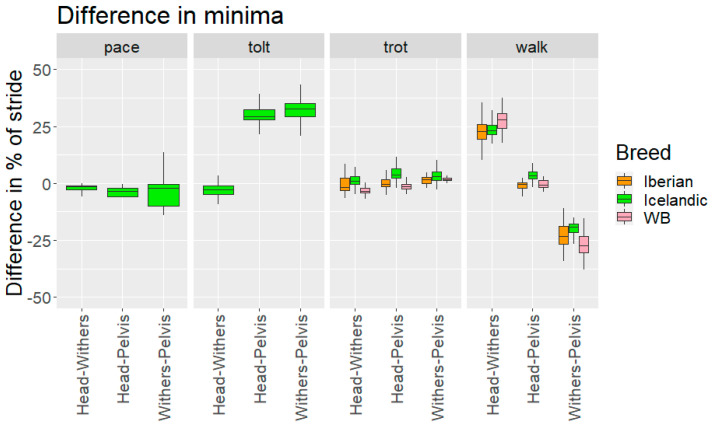
Timing difference in vertical displacement minima between head, withers and pelvis for the different breeds and gaits studied, based on LH hoof-on stride split. WB: Warmblood horses. A positive Head–Withers value indicates that the head reaches the minima first. A positive Head–Pelvis means the head reaches the minima first. A positive Withers–Pelvis means the withers reaches the minima first.

**Figure 8 animals-12-03053-f008:**
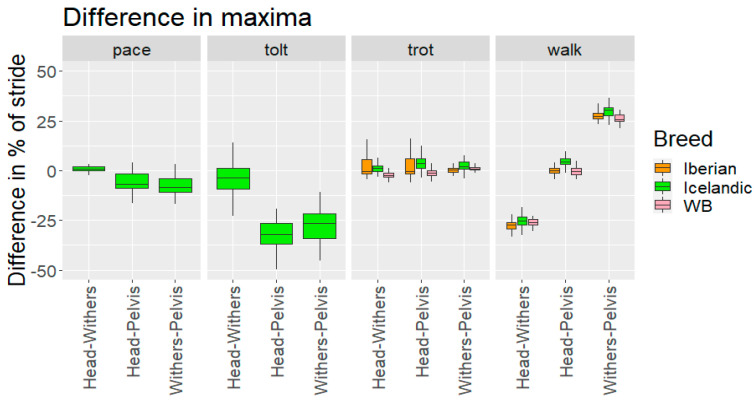
Timing difference in vertical displacement maxima between head, withers and pelvis for the different breeds and gaits studied. WB: Warmblood horses. A positive Head–Withers value indicates that the head reaches the minima first. A positive Head–Pelvis means the head reaches the minima first. A positive Withers–Pelvis means the withers reaches the minima first.

**Table 1 animals-12-03053-t001:** Stride parameters (median and interquartile range (IQR)) for each breed, gait and condition (ridden/in hand). N: number of horses in each category, WB: Warmblood horses. * as % of stride duration; FL: forelimbs, HL: hind limbs. Diagonal dissociation was not calculated for pace and tölt.

	Pace	Tölt	Trot				Walk			
	Icelandic Ridden	Icelandic Ridden	Iberian	Icelandic In Hand	Icelandic Ridden	Warmblood	Iberian	Icelandic In Hand	Icelandic Ridden	Warmblood
Stride duration (s)										
Median (IQR)	0.42 (±0.029)	0.55 (±0.068)	0.74 (±0.037)	0.63 (±0.083)	0.59 (±0.085)	0.78 (±0.048)	1.2 (±0.15)	1.0 (±0.20)	0.99 (±0.13)	1.3 (±0.11)
Diagonal dissociation *										
Median (IQR)			−1.2 (±3.5)	−4.8 (±3.3)	−5.7 (±5.4)	−2.6 (±4.0)	−23 (±3.0)	−26 (±4.0)	−25 (±2.7)	−25 (±2.5)
Lateral dissociation *										
Median (IQR)	9.7 (±4.3)	18 (±7.7)	49 (±3.6)	45 (±2.9)	44 (±5.5)	47 (±4.2)	27 (±2.9)	23 (±4.0)	25 (±2.4)	25 (±2.1)
Suspension *										
Median (IQR)	13 (±7.6)	0 (±0)	2.1 (±4.2)	3.7 (±4.8)	0.72 (±3.2)	4.0 (±5.4)	0 (±0)	0 (±0)	0 (±0)	0 (±0)
Stance Duration FL (s)										
Median (IQR)	0.15 (±0.018)	0.25 (±0.061)	0.37 (±0.051)	0.30 (±0.052)	0.27 (±0.060)	0.38 (±0.042)	0.77 (±0.13)	0.63 (±0.13)	0.63 (±0.11)	0.85 (±0.090)
Stance Duration HL (s)										
Median (IQR)	0.17 (±0.0080)	0.25 (±0.054)	0.32 (±0.042)	0.27 (±0.028)	0.28 (±0.047)	0.31 (±0.040)	0.71 (±0.12)	0.58 (±0.14)	0.59 (±0.075)	0.74 (±0.077)
Swing Duration FL (s)										
Median (IQR)	0.28 (±0.022)	0.30 (±0.023)	0.37 (±0.040)	0.34 (±0.032)	0.31 (±0.030)	0.39 (±0.032)	0.43 (±0.050)	0.38 (±0.053)	0.37 (±0.041)	0.45 (±0.037)
Swing Duration HL (s)										
Median (IQR)	0.26 (±0.027)	0.30 (±0.027)	0.42 (±0.042)	0.37 (±0.047)	0.32 (±0.039)	0.46 (±0.038)	0.48 (±0.046)	0.43 (±0.076)	0.41 (±0.049)	0.55 (±0.038)
Duty factor FL										
Median (IQR)	0.36 (±0.022)	0.46 (±0.059)	0.49 (±0.060)	0.46 (±0.048)	0.46 (±0.053)	0.49 (±0.038)	0.64 (±0.032)	0.63 (±0.022)	0.62 (±0.025)	0.65 (±0.026)
Duty factor HL										
Median (IQR)	0.40 (±0.024)	0.45 (±0.049)	0.43 (±0.055)	0.42 (±0.032)	0.46 (±0.049)	0.39 (±0.039)	0.59 (±0.033)	0.58 (±0.022)	0.59 (±0.026)	0.57 (±0.022)

**Table 2 animals-12-03053-t002:** Vertical range of motion (mm) at walk in hand for head, withers and pelvis. Results presented are estimated marginal means (EMM) with confidence intervals (CI) from the linear mixed model with pairwise comparisons. n = number of strides.

	Head			Withers			Pelvis			Within-Breed Significant Pairwise Comparison (*p* < 0.01)
	EMM	Lower CI	Upper CI	EMM	Lower CI	Upper CI	EMM	Lower CI	Upper CI	
**Iberian** **(n = 1090)**	104.7	95.3	114	26.8	17.4	36.2	68	58.6	77.3	All
**Icelandic** **(n = 837)**	79.3	71.3	87.4	36	28	44	65.8	57.8	73.8	All
**WB** **(n = 1000)**	117.1	107	127.2	33.4	23.3	43.5	78.8	68.7	88.9	All
**Between-breed significant pairwise comparison** **(*p* < 0.001)**	All			None			None			

**Table 3 animals-12-03053-t003:** Vertical range of motion (mm) at trot in hand for head (H), withers (W) and pelvis (P). Values presented are estimated marginal means (EMM) with confidence intervals (CI) from the linear mixed model with pairwise comparisons. n = number of strides.

	Head			Withers			Pelvis			Within-Breed Significant Pairwise Comparison (*p* < 0.01)
	EMM	Lower CI	Upper CI	EMM	Lower CI	Upper CI	EMM	Lower CI	Upper CI	
**Iberian** **(n = 1540)**	63.7	56.8	70.6	66.2	59.3	73.1	70.8	63.9	77.7	None
**Icelandic** **(n = 815)**	60.7	54.7	66.7	46.8	40.8	52.9	56	50	62	H-W, W-P
**WB** **(n = 1340)**	78.1	70.5	85.7	92.2	84.6	99.8	89.1	81.5	96.7	H-W, H-P
**Between-breed significant pairwise comparison (*p* < 0.001)**	Icelandic–Warmblood Iberian–Warmblood			All			All			

**Table 4 animals-12-03053-t004:** Timing of minima and maxima. Values for head and withers expressed as percentage of left forelimb stance and values for pelvis expressed as a percentage of left hindlimb stance. Statistically significant differences between breeds are indicated separately for minima and maxima by pairs of values with the same letters (A–G), with level of significance indicated by asterisks (* *p* < 0.05, ** *p* < 0.01, *** *p* < 0.001).

		Minima Median (±Interquartile Range)				Maxima Median (±Interquartile Range)			
		Iberian	Warmblood	Icelandic	Icelandic	Iberian	Warmblood	Icelandic	Icelandic
		In hand	In hand	In hand	Ridden	In hand	In hand	In hand	Ridden
**Head**	**Walk**	48 (±5.4)	47 (±5.5)	46 (±4.6)	46 (±5.5)	88 (±5.6)	87 (±5.5)	86 (±3.6)	88 (±4.7)
	**Trot**	41 (±9.4)	46 (±7.9)	39 (±5.2)	40 (±13)	92 (±20) D **/E **	97 (±6.6)	92 (±9.1) D **	94 (±11) E **
	**Tölt**				49 (±15)				4.8 (±20)
	**Pace**				27 (±12)				95 (±11)
**Withers**	**Walk**	5.3 (±14) A **/B *	13 (±11) A **	4.6 (±5.3) B *	3.7 (±4.7)	42 (±6.6) D **	44 (±4.8)	48 (±5.3) D **	45 (±6.3)
	**Trot**	39 (±5.0)	40 (±4.4)	42 (±5.6)	41 (±9.8)	91 (±6.7) D **	92 (±4.6) E *	94 (±5.6)	98 (±7.5) D **/E *
	**Tölt**				39 (±13)				−0.31 (±24)
	**Pace**				27 (±9.6)				99 (±14)
**Pelvis**	**Walk**	10 (±2.5) A **	7.7 (±2.6) A **/B **/C **	11 (±2.4) B ***	11 (±2.5) C ***	55 (±44) D **	51 (±3.3) D **/E **/F **	57 (±3.8) E **	56 (±6.1) F **
	**Trot**	45 (±7.1)	42 (±6.0)	41 (±6/7)	37 (±5.5)	100 (±5.5) D ***/E **	110 (±3.8) E **/F **/G ***	100 (±6.2) F **	92 (±5.2) D ***/G ***
	**Tölt**				40 (±8.7)				97 (±7.1)
	**Pace**				41 (±13)				100 (±17)

**Table 5 animals-12-03053-t005:** Difference in timing of vertical minima and maxima between different axial body segments (head–withers, head–pelvis, withers–pelvis) in Iberian, Warmblood, and Icelandic horses performing different gaits. Significant differences in pair-wise comparisons between breeds are indicated by the same letters (A, B) and asterisks indicating the level of significance (* *p* < 0.05, ** *p* < 0.01, *** *p* < 0.001).

		Difference in Minima as % of Stride Median (±Interquartile Range)			Difference in Maxima as % of Stride Median (±Interquartile Range)		
		Iberian	Warmblood	Icelandic	Iberian	Warmblood	Icelandic
**Head-Withers**	**Walk**	23 (±6.7) A ***	28 (±6.3) A ***	23 (±4.4)	−27 (±3.4) A **	−26 (±2.6)	−25 (±3.8) A **
	**Trot**	−1.6 (±5.4) A ***	−3.3 (±2.2) A ***/B **	1.1 (±3.3) B **	−0.53 (±7.7) A ***	−2.3 (±2.2) A ***	1 (±2.7)
	**Tölt**			−2.9 (±3.9)			−3.6 (±10)
	**Pace**			−1.3 (±1.9)			0.9 (±2.2)
**Head-Pelvis**	**Walk**	−0.5 (±2.6) A **	−0.7 (±2.9)	3.3 (±3.3) A **	−0.2 (±2.2) A **	−0.4 (±3.1) B **	4.3 (±3.1) A **/B **
	**Trot**	−0.3 (±3.0)	−1.4 (±2)	3.8 (±4.3)	−0.3 (±8.0)	−1.4 (±2.5) B **	3.7 (±4.7) B **
	**Tölt**			29 (±4.7)			−32 (±11)
	**Pace**			−3.4 (±3.8)			−7 (±7.0)
**Withers-Pelvis**	**Walk**	−23 (±7.8) A *	−28 (±7.6) A */B ***	−19 (±3.7) B ***	27 (±3.2)	26 (±3.1)	30 (±3.7)
	**Trot**	1.6 (±2.7) A **	1.8 (±1)	3 (±3.7) A **	0.2 (±1.8)	0.7 (±1.4)	2 (±3.4)
	**Tölt**			33 (±5.9)			−26 (±12)
	**Pace**			−2.2 (±9.6)			−8.5 (±6.8)

## Data Availability

The data that support the findings of this study are available from the corresponding author upon reasonable request.

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
