# Peer review of "Timing of Vertical Head, Withers and Pelvis Movements Relative to the Footfalls in Different Equine Gaits and Breeds"

_animals, 2022, doi:10.3390/ani12213053_

Round 1
Reviewer 1 Report
The manuscript is very interesting and covers many new aspects of horse locomotion and its evaluation. The main problem with the manuscript is connected with data presentation and structure.
The results section is lacking (there are only figures and tables not cited in the text), instead of that, the discussion part consists of some results descriptions and a basic and short discussion. That is why I would suggest a major revision of the paper. The Authors should also take into account that Animals is read by a broad audience so the language shortcuts coming from biomechanics are not proper.
Please find enclosed pdf.

Author Response
Thank you for all the valuable suggestions to improve the manuscript.
Please see the attachment for the point-by-point response.

Reviewer 2 Report
Overview and general recommendation:
I think this is an important piece of work documenting several key parameters of horses’ gaits at a walk and tolt, and in a variety of other breeds. The paper presents a wealth of information which I thought was presented well. I thought the introduction was thorough and provides a suitable level of detail for a reader who may not be well versed on the topic at hand.
Section 3.6, the results on timing of the vertical movement of head, withers and pelvis is presented in figures for each subsection (gait) and tables with the text not saying much about the figures and tables, except to point out what they are. I do wonder how this section could be better presented for publication i.e. whether this could be moved to the supplementary or otherwise. Having said this, as a visual learner, I do highly appreciate the manner the data was presented, so I will defer to the editor’s thoughts on this section.
Specific comments
Introduction
1. Lines 47-48 – Consider rewording “objectively-based signs of lameness”.
2. Line 140 – “A prerequisite to the use of..”
Results
1. Line 238 - Table 1 title. Are the values in the table median and IQR as stated in the table and methods? The title says “medial and SD”.
2. Table 1 – Sorry if I have missed something in the methods and/or text with this query, but can you clarify the number of horses in each category in Table 1? Was this carried out on a larger set of horses than your final dataset of 23 Icelandic, 19 WB, and 23 Iberian horses, before application of the selection criteria?
3. Line 277-278 – Should this be reference Table 3 instead?
4. Figures 3 and 4 – The x-axis labelling of “Stride normalised” is missing for Head (but present for Figures 5 and 6).
5. Figures 5 and 6 – This is less important, and I leave it up to the authors to action or not – but it would be nice for the colours within this figure to match the colours for Icelandic, ridden horses of Figures 3 and 4.
6. Line 323 – Should this be referencing Table 5 instead?
Author Response

(The authors gave the same response as above.)

Reviewer 3 Report
General comments
Summary is not present.
Introduction
Sections 1.1 and 1.3: please add appropriate references were missing through the section (see lines 67-69 and 69-71, 86-88).
Line 101: “it” needs a capital letter. Please check grammar errors in the text.
Materials and Methods
Section 2.1.2: which kind of horses were included? Were they leisure or competitive horses? How often were they trained during a week? If proper, which kind of competition have they been trained for? Have they followed a period of IMU sensors adaptation before the investigation started?
Section 2.1.2: Were data sets chosen according to manufacture indication? Please add comments. Were data sets used as inclusion criteria? Please clarify.
Section 2.1.2: which were data collection conditions? Were the horses walked and trotted by the same operator? Which kind of surfaces were used? Please clarify.
Lines 168-169: It would be useful for the reader to add a figure to illustrate the sensor positioning on the horse. Please consider adding it.
Results
Line 231: “centimetres” (cm) is missing.
Table 1: the present table is hard to read and fully understand. Significative differences in results were not highlighted. Consider splitting and revising it.
Tables: I found difficulties in matching results described in the main text with their own table. It seems that tables do not follow the numerical order, and thus the results are less readable. Please, check and revise it.
Discussion
Lines 342-343: conditions of sensors application and measurements have not been described. Thus, it would be necessary to deeply describe data collection methods and then comment on them.
Lines 350-351: the present phrase is similar to the previous one (lines: 341-342). Consider to rephrase.
Lines 366-368: The horse’s stride is also affected on front and hind limb upper limb angles. Would it influence the present results? Please, add comments.
Lines 382-384: Please add information regarding the data collection environment (see material and methods section) and then draw consideration in the paper discussion. Would different surfaces used be considered a limitation of the present study?
Author Response

(The authors gave the same response as above.)

Reviewer 4 Report
While understanding of the normal locomotion of various horse breeds is important to the clinician in order to detect lameness, the aim of this study is not clear and in it's current state much of the information is not novel to what is currently published. Although the title indicates the study will focus on various breeds, there are only three breeds studied in which all three of these breeds have been well-researched particularly concerning the variables measured. In fact, the temporal variable measurements for these breeds and their associated gaits have been published well over 20 years ago, especially concerning the Warmblood. Why were these breeds selected, besides just for convenience? It seems odd to compare a small pony breed to two much taller breeds when looking at vertical movement. Breeds of similar height would seem more appropriate for this study. In addition, the gaits have been well-studied, even the less known gaits of the tolt and pace. Although much of the research on the pace historically has utilized Standardbreds instead of the Icelandic horse, the basic measurements performed within this study lack true novelty for publication. Along with justification for why the particular breeds were selected, it's also confusing and lacks justification as to why only one breed was ridden and not the other two breeds. Nevertheless, the use of the IMU sensors offers a unique perspective, and thus, the study could be directed more into focusing on the use of new, more accessible technology in lameness detection, however, the introduction and even the title do not indicate this as a focus of the study. This aspect of the study is further lacking within the discussion and conclusions sections of the manuscript. In addition, if the utilization of the sensors is the focus of the study, then, the IMU sensor data should have been compared to other assessment methods and potentially lame horses should have been utilized within the study. Simply, there doesn't appear to be any clear, concise objectives to the study and the study never gives any specific hypothesis to reflect a focus of the work being done. In the end, direction is needed and justification as to why this direction was selected should be clear.
As for the introduction, this too adds confusion as to the focus of the study. While the majority of the introduction focuses on gaited breeds and how their gaits are uniquely different from other gaits, there is only one out of the three breeds that would be considered gaited and only one gait studied within this manuscript that would be classified as uniquely a gaited horse gait. Furthermore, the focus is on comparing gaits, and thus, why aren't the gaits more directly compared within the introduction? In other words, why are the definitions and descriptions of the gaits divided up into different subsections, rather than compared within the same paragraph? In addition, there are numerous studies done on all of these gaits and on these three breeds, and yet, the references are quite limited. Furthermore, some of the more accepted and more thoroughly utilized terminology when it comes to gaited breeds and describing, defining their respective gaits is not utilized. This further suggests a lack of potentially reviewed literature. As for the second half of the introduction, the focus is on lameness, and yet, there were no horses studied that were lame. The section also brings up the discussion of ambling gaits, and yet, within the sections of the introduction focused on defining gaits, the ambling gait was never defined. In any case, without data collected specifically on lame horses, this section directed on lameness should be moved within the discussion of gaits to make comparisons between "normal" versus "abnormal" and help to further define the gaits. Finally, within the introduction the authors need to explain the following giving further justification: why were these specific three breeds studied and compared, especially when only one was a pony breed; why two were non-gaited breeds and only one was gaited; and why were horses both led and ridden and why was that only done for one breed?
As for the methods section, it needs to clearly begin with a statement concerning the humane and ethical use of animals for research and that this study was reviewed and approved by such a committee within each of the countries that the research took place. Each country has it's own standards concerning animal research, and thus, a statement is necessary concerning that this research was reviewed and approved by a committee charged with assessing the humane and ethical use of animals for research specific to each country's standards. If this type of review was not done within each of the countries where data was collected prior to data collection, then, this data cannot be utilized. As for the horses, how and when was soundness determined and who was qualified to determine soundness? Even the authors state within the introduction that there is a need for objective assessment methods for quantifying lameness in gaits besides the trot, and thus, it is concerning if lameness evaluation did not include diagnostic imaging tools to rule out potential lamenesses. As for horse selection for this study, how were horses selected? Do these horses represent the "ideal" for their breed and would training have some type of influence on gait performance? Research has indicated that not only breed type can result in variations within the trot, but also performance type, particularly associated with training, can influence the trot, and thus, how were these variables controlled for when selecting the sample population? Along with variability associated with training within each of the breeds, there is potential of variability due to the riders/handlers utilized unless the same rider/handler were used for all of the horses analyzed. What makes these riders/handlers qualified for presenting these gaits? In addition, was ground surfaces controlled for and was velocity controlled for as both variables can also influence the gait variables measured within this study?
As for the results and discussion section, the information concerning temporal variables within both sections is redundant from what has been done in previous studies and has been well documented. This information should only be reported and discussed as it relates to what data was collected from the IMU sensors concerning displacements. Furthermore, the temporal variable data can all be related back to influences concerning both the uniqueness in the conformation of each of the three breeds and in the velocities of which the gaits were performed, thus, without correcting for height differences and controlling the velocity of the gaits during data collection, the discussion concerning temporal variables is flawed. Even the authors address this limitation within the discussion, and while it is pointed out, it is a significant weakness of this study limiting the value of what can be taken from the information that is given at this time. On a final note, there are too many tables and figures for what data is novel specifically to this study and many of these tables and figures are hard to read, and thus, by removing some of these tables and figures, others can be enlarged and adjusted so that they can be better reviewed. Furthermore, for both of the results and discussion, similar to the introduction, it would be helpful to minimize the use of subsections and to try to merge together sections so that the authors relate how variables interact and influence each other. The reporting, at this point, is fairly simplistic and lacks in depth interpretation of how the variables work together to perform the gaits being studied. It is within this interaction between the various mechanics of the gait that we can see how the horse adapts to and compensates for lamenesses. Understanding this interaction and the usefulness of new sensor technology for detecting this interaction when it becomes flawed or abnormal gait would be of value to journal readers.
Author Response

(The authors gave the same response as above.)

Round 2
Reviewer 1 Report
animals-1860504 - Revised Review
The paper has been improved significantly. However, there are still some points that should be corrected for better clearance and correct manuscript structure. In general, the manuscript needs another careful review – the results section is not improved enough and some mistakes are coming from less patience. For example different information in simple summary and abstract regarding results once “for all gaits except walk and pace” and then “for all gaits except walk” without pace. The result section is still the weakest part of the manuscript. The authors are not summarizing their results - they just let the readers look at the graphs that are too small in some parts and not visible enough. It is not the role of the readers to guess what differences can be seen in the figures.
In detail:
Simple summary – half of the text discusses lameness which is not the main subject of the paper. One, or two sentences are enough as you did not study lameness directly in this paper.
Abstract - L 28 – the start of the suspension of the stride cycle – that is not detailed information – 1. the stride can start from every limb, 2. In trot in the front pair - one limb is on the ground, the other is in the swing phase – so for one limb, it will be the start of suspension, for another full contact. That is not clear.
L 34 – wording - the relationship is usually connected with calculations of correlations that you did not investigate
L 36 – why warmblood is by the small letter?
L 26 and 41 – not the same result is presented
Keywords – lameness is not investigated in this paper, perhaps – lameness evaluation would be better (even not perfect)?
Introduction – it is better written and connected with results achieved, however some phrases too specialist
L 54 – pendulum and spring-mass mechanism – please extend the phrases (perhaps more sentences will be needed)
L 59 – using the same marks of breeds will be easier for the reader – in the paper once you use shortcuts, the other time full names – please uniform it
L 62 – the amble is introduced but running/walking gaits from L 99, and 101 are not. Please correct.
L 69-70 –add for comparison in this sentence.
L 80-81 this is clear when you take into account one limb, what about pairs of legs in symmetrical gaits? Please address it in the paper. It must be clear for all.
L 84 – CoM – shortcut not introduced earlier
Material and methods
L 133-134 – please avoid a one-sentence paragraph. It is also not clear what you mean by “sound horses.”
L 136-138 – it should be moved to section 2.1
L 137 – you should be very clear in the selection of the data you use – you write about it throughout the paper and it is not clear if it is one or three steps of data selection. It is not correct that the accuracy of your selection is given in the discussion – all steps of data selection should be clearly stated as a separate paragraph (with stride normalization that you cited only). Please provide a separate paragraph on the selection of the data with all steps of selection in one part of M&M.
L 139 – horses are fine described, however, the first sentence should be more general without “;”. For example “the final data set consisted of three groups. Twenty …”
You have a lot of data – did you think about the calculation of repeatability for these three groups of horses? That would be interesting as well.
Figure 1 – it is not clear if the accuracy you discuss in the discussion (80%) is the one you mention in this figure (HD/PD). As stated above please add the special section on data selection.
L 155 – different sampling rate has to be discussed in the limitation part, it probably might influence the result; if not according to your knowledge - discuss it also.
L 173 – log files – I am not sure what you mean (is it significant for the whole study?)
L 176- over-air-reconfiguration – it is not clear and I do not know if it is important at all for your study
L 178 – later data checks – all data selection and checks should be written clearly in one separate section
L 180-189-195 paragraph with all parameters would be better; they have to be visible through the paper to let the reader come back if needed
L 189 – please give more detail on this normalization (in short)
L 196-198 should be in the section on data selection (all should be in one piece for better clearance). This aspect of data selection should be mentioned in the limitation part. It is possible that thanks to the reduction of cases you have changed the variability of gait parameters. Please discuss its possible influence on the results in the limitation part. (for example for a p-value that was 0.06)
L 206 – it is not clear what it means “in relation to stance phase”. Add “position” to the sentence. Is it a new description/trait/parameter? Write it in the section “parameters”. Perhaps the stride normalization description (in short as mentioned above 189) would solve this unclear meaning.
Please discuss in the limitation part that the breed effect is biased by the investigation center.
L 215 – please check if these shortcuts are present in each of your table legend
Results
Table 1 – L 221/222 – such information should be clearly stated in the section on parameters measured (to be done). It may not appear for the first time in the table legend.
Please provide the information in this planned section - why you do not have suspensions for a walk and toelt/ in contrast to the lack of data for DD in table 1.
L 225 – why only walk and trot are reported?
Perhaps one sentence on pace/toelt should be given – they were investigated as well.
L 223 – suspension time – give gaits in the title – you should write in detail what are you writing about. Add trot/pace in the title and report something on all your results
L 235 – 0.06 is not borderline significant, it is out of the significance – please correct at least for “close to significance”
L 243 – I do not understand why you do not report (in shortcuts) all your data. Why it is only trot?
Write about the main observed information from all studied traits/groups.
Reviewer 3 Report
Manuscript Number: animals- 1860504
Title: Vertical movement of head, withers and pelvis, and their timing relative to footfall, in different equine gaits and breeds
Materials and Methods
Line 132: layout errors present. Please, correct it.
Section 2.1.2: it would be necessary to add your clarification to the main text. You could find the review 1 question, which I refer to, following: “which kind of horses were included? Were they leisure or competitive horses? How often were they trained during a week? If proper, which kind of competition have they been trained for? Have they followed a period of IMU sensors adaptation before the investigation started? “
Section 2.1.2: it would be necessary to add your clarification to the main text. You could find the review 1 question, which I refer to, following: “Were data sets chosen according to manufacture indication? Please add comments. Were data sets used as inclusion criteria? Please clarify. “
Discussion
Lines 366-368: it would be necessary to add your clarification to the main text, and discuss it as a possible limitation. You could find the review 1 question, which I refer to, following: “The horse’s stride is also affected on front and hind limb upper limb angles. Would it influence the present results? Please, add comments.”
Lines 382-384: it would be necessary to add your clarification to the main text, and discuss it as a possible limitation. You could find the review 1 question, which I refer to, following: “Please add information regarding data collection environment (see material and methods section) and then draw consideration in paper discussion. Would different surfaces used be considered a limitation of the present study?"
Reviewer 4 Report
While the authors should be commended on what revisions that were made, there are still suggested revisions presented by multiple reviewers that were not addressed and that is due to the limitation of the study itself and how it was performed. Sadly, without completely redoing aspects of the study, it is hard to rectify some of the errors that occurred that jeopardize the scientific soundness of the work that is done. Refer to previous reviewers' suggestions for discussion concerning some of these limitations. Outside of the limitations to the study itself, further exploration into the background of this research needs to be attempted within the introduction. The complexity of the study and the multiple directions of what is trying to be studied is not fully explored within the introduction and the introduction itself is hard to follow. The same can be said concerning the discussion section. It is hard to follow at times and does not represent a deeper understanding of the work that was done and how it relates to previous research. Discussion concerning temporal variables and the timing of the limbs has been well documented in multiple previous studies so what makes this research unique? This truly was not explored nor emphasized within the discussion and conclusions.
Round 3
Reviewer 1 Report
The answer of the Authors to the sampling rate subject in the limitation part (L155 earlier version) is very interesting and very informative.
It should be added in the method part (better) or limitation part as lack of limitation. Please add all information to the paper. The paper should be clear and understandable for all readers, also no specialists.
Author Response
Thanks for the comment. In the previously submitted version we already clearly state the sampling rate in the Material and methods line 168-169. It is not clear why reviewer is asking to add this as limitation. We did not make any changes to the current resubmitted manuscript as this information was already presented.
Reviewer 3 Report
Manuscript Number: animals- 1860504
Title: Vertical movement of head, withers and pelvis, and their timing relative to footfall, in different equine gaits and breeds
4.5. Limitations
You should support your consideration with litteratur references, please rvise all th epresetn paragraphs and add appropriate references where required for your study’s limits consideration
Author Response
Thanks for the comment. We have added references to our limitations when appropriate and relevant evidence is available.